# VARIATIONAL BAYESIAN PHYLOGENETIC INFERENCE

**Cheng Zhang, Frederick A. Matsen IV**
Computational Biology Program
Fred Hutchinson Cancer Research Center
Seattle, WA 98109, USA
`{czhang23,matsen}@fredhutch.org`

## ABSTRACT

Bayesian phylogenetic inference is currently done via Markov chain Monte Carlo with simple mechanisms for proposing new states, which hinders exploration efficiency and often requires long runs to deliver accurate posterior estimates. In this paper we present an alternative approach: a variational framework for Bayesian phylogenetic analysis. We approximate the true posterior using an expressive graphical model for tree distributions, called a subsplit Bayesian network, together with appropriate branch length distributions. We train the variational approximation via stochastic gradient ascent and adopt multi-sample based gradient estimators for different latent variables separately to handle the composite latent space of phylogenetic models. We show that our structured variational approximations are flexible enough to provide comparable posterior estimation to MCMC, while requiring less computation due to a more efficient tree exploration mechanism enabled by variational inference. Moreover, the variational approximations can be readily used for further statistical analysis such as marginal likelihood estimation for model comparison via importance sampling. Experiments on both synthetic data and real data Bayesian phylogenetic inference problems demonstrate the effectiveness and efficiency of our methods.

## 1 INTRODUCTION

Bayesian phylogenetic inference is an essential tool in modern evolutionary biology. Given an alignment of nucleotide or amino acid sequences and appropriate prior distributions, Bayesian methods provide principled ways to assess the phylogenetic uncertainty by positing and approximating a posterior distribution on phylogenetic trees (Huelsenbeck et al., 2001). In addition to uncertainty quantification, Bayesian methods enable integrating out tree uncertainty in order to get more confident estimates of parameters of interest, such as factors in the transmission of Ebolavirus (Dudas et al., 2017). Bayesian methods also allow complex substitution models (Lartillot & Philippe, 2004), which are important in elucidating deep phylogenetic relationships (Feuda et al., 2017).

Ever since its introduction to the phylogenetic community in the 1990s, Bayesian phylogenetic inference has been dominated by random-walk Markov chain Monte Carlo (MCMC) approaches (Yang & Rannala, 1997; Mau et al., 1999; Huelsenbeck & Ronquist, 2001). However, this approach is fundamentally limited by the complexities of tree space. A typical MCMC method for phylogenetic inference involves two steps in each iteration: first, a new tree is proposed by randomly perturbing the current tree, and second, the tree is accepted or rejected according to the Metropolis-Hastings acceptance probability. Any such random walk algorithm faces obstacles in the phylogenetic case, in which the high-posterior trees are a tiny fraction of the combinatorially exploding number of trees. Thus, major modifications of trees are likely to be rejected, restricting MCMC tree movement to local modifications that may have difficulty moving between multiple peaks in the posterior distribution (Whidden & Matsen IV, 2015). Although recent MCMC methods for distributions on Euclidean space use intelligent proposal mechanisms such as Hamiltonian Monte Carlo (Neal, 2011), it is not straightforward to extend such algorithms to the composite structure of tree space, which includes both tree topology (discrete object) and branch lengths (continuous positive vector) (Dinh et al., 2017).

Variational inference (VI) is an alternative approximate inference method for Bayesian analysis which is gaining in popularity (Jordan et al., 1999; Wainwright & Jordan, 2008; Blei et al., 2017). Unlike MCMC methods that sample from the posterior, VI selects the best candidate from a family of tractable distributions to minimize a statistical distance measure to the target posterior, usually the Kullback-Leibler (KL) divergence. By reformulating the inference problem into an optimization problem, VI tends to be faster and easier to scale to large data (via stochastic gradient descent) (Blei et al., 2017). However, VI can also introduce a large bias if the variational distribution is insufficiently flexible. The success of variational methods, therefore, relies on having appropriate tractable variational distributions and efficient training procedures.

To our knowledge, there have been no previous variational formulations of Bayesian phylogenetic inference. This has been due to the lack of an appropriate family of approximating distributions on phylogenetic trees. However the prospects for variational inference have changed recently with the introduction of *subsplit Bayesian networks* (SBNs) (Zhang & Matsen IV, 2018), which provide a family of flexible distributions on tree topologies (i.e. trees without branch lengths). SBNs build on previous work (Höhna & Drummond, 2012; Larget, 2013), but in contrast to these previous efforts, SBNs are sufficiently flexible for real Bayesian phylogenetic posteriors (Zhang & Matsen IV, 2018).

In this paper, we develop a general variational inference framework for Bayesian phylogenetics. We show that SBNs, when combined with appropriate approximations for the branch length distribution, can provide flexible variational approximations over the joint latent space of phylogenetic trees with branch lengths. We use recently-proposed unbiased gradient estimators for the discrete and continuous components separately to enable efficient stochastic gradient ascent. We also leverage the similarity of local structures among trees to reduce the complexity of the variational parameterization for the branch length distributions and provide an extension to better capture the between-tree variation. Finally, we demonstrate the effectiveness and efficiency of our methods on both synthetic data and a benchmark of challenging real data Bayesian phylogenetic inference problems.

## 2 BACKGROUND

**Phylogenetic Posterior**   A phylogenetic tree is described by a tree topology $\tau$ and associated non-negative branch lengths $\boldsymbol{q}$. The tree topology $\tau$ represents the evolutionary diversification of the species. It is a bifurcating tree with $N$ leaves, each of which has a label corresponding to one of the observed species. The internal nodes of $\tau$ represent the unobserved characters (e.g. DNA bases) of the ancestral species. A continuous-time Markov model is often used to describe the transition probabilities of the characters along the branches of the tree. Let $\boldsymbol{Y} = \{Y_1, Y_2, \ldots, Y_M\} \in \Omega^{N \times M}$ be the observed sequences (with characters in $\Omega$) of length $M$ over $N$ species. The probability of each site observation $Y_i$ is defined as the marginal distribution over the leaves

$$p(Y_i|\tau, \boldsymbol{q}) = \sum_{a^i} \eta(a^i_\rho) \prod_{(u,v) \in E(\tau)} P_{a^i_u a^i_v}(q_{uv}) \tag{1}$$

where $\rho$ is the root node (or any internal node if the tree is unrooted and the Markov model is time reversible), $a^i$ ranges over all extensions of $Y_i$ to the internal nodes with $a^i_u$ being the assigned character of node $u$, $E(\tau)$ denotes the set of edges of $\tau$, $P_{ij}(t)$ denotes the transition probability from character $i$ to character $j$ across an edge of length $t$ and $\eta$ is the stationary distribution of the Markov model. Assuming different sites are identically distributed and evolve independently, the likelihood of observing the entire sequence set $\boldsymbol{Y}$ is $p(\boldsymbol{Y}|\tau, \boldsymbol{q}) = \prod_{i=1}^M p(Y_i|\tau, \boldsymbol{q})$. The phylogenetic likelihood for each site in equation 1 can be evaluated efficiently through the pruning algorithm (Felsenstein, 2003), also known as the sum-product algorithm in probabilistic graphical models (Strimmer & Moulton, 2000; Koller & Friedman, 2009; Höhna et al., 2014). Given a proper prior distribution with density $p(\tau, \boldsymbol{q})$ imposed on the tree topologies and the branch lengths, the phylogenetic posterior $p(\tau, \boldsymbol{q}|\boldsymbol{Y})$ is proportional to the joint density

$$p(\tau, \boldsymbol{q}|\boldsymbol{Y}) = \frac{p(\boldsymbol{Y}|\tau, \boldsymbol{q})p(\tau, \boldsymbol{q})}{p(\boldsymbol{Y})} \propto p(\boldsymbol{Y}|\tau, \boldsymbol{q})p(\tau, \boldsymbol{q})$$

where $p(\boldsymbol{Y})$ is the intractable normalizing constant.

**Subsplit Bayesian Networks**   We now review subsplit Bayesian networks (Zhang & Matsen IV, 2018) and the flexible distributions on tree topologies they provide. Let $\mathcal{X}$ be the set of leaf labels.

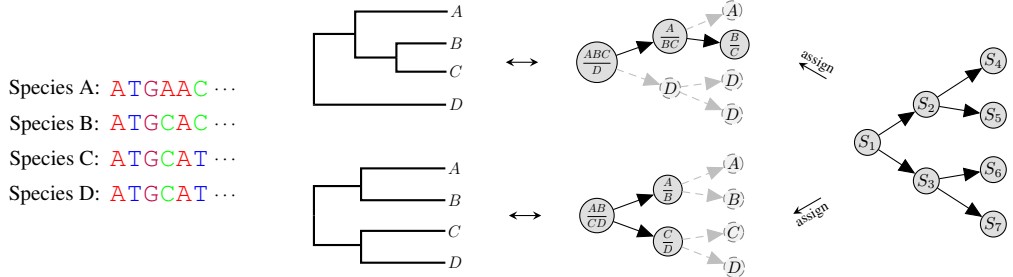

Figure 1: A simple subsplit Bayesian network for a leaf set that contains 4 species. **Left**: A leaf label set $\mathcal{X}$ of 4 species, each label corresponds to a DNA sequence. **Middle (left)**: Examples of (rooted) phylogenetic trees that are hypothesized to model the evolutionary history of the species. **Middle (right)**: The corresponding SBN assignments for the trees. For ease of illustration, subsplit $(W, Z)$ is represented as $\frac{W}{Z}$ in the graph. The *dashed gray subgraphs* represent fake splitting processes where splits are deterministically assigned, and are used purely to complement the networks such that the overall network has a fixed structure. **Right**: The SBN for these examples.

We call a nonempty subset of $\mathcal{X}$ a *clade*. Let $\succ$ be a total order on clades (e.g., lexicographical order). A *subsplit* $(W, Z)$ of a clade $X$ is an ordered pair of disjoint subclades of $X$ such that $W \cup Z = X$ and $W \succ Z$. A subsplit Bayesian network $B_{\mathcal{X}}$ on a leaf set $\mathcal{X}$ of size $N$ is a Bayesian network whose nodes take on subsplit or singleton clade values that represent the local topological structure of trees (Figure 1). Following the splitting processes (see the *solid dark subgraphs* in Figure 1, middle right), rooted trees have unique subsplit decompositions and hence can be uniquely represented as compatible SBN assignments. Given the subsplit decomposition of a rooted tree $\tau = \{s_1, s_2, \ldots\}$, where $s_1$ is the root subsplit and $\{s_i\}_{i>1}$ are other subsplits, the SBN tree probability is

$$p_{\mathrm{sbn}}(T = \tau) = p(S_1 = s_1) \prod_{i>1} p(S_i = s_i | S_{\pi_i} = s_{\pi_i})$$

where $S_i$ denotes the subsplit- or singleton-clade-valued random variables at node $i$ and $\pi_i$ is the index set of the parents of $S_i$. The Bayesian network formulation of SBNs enjoys many benefits: i) *flexibility*. The expressiveness of SBNs is freely adjustable by changing the dependency structures between nodes, allowing for a wide range of flexible distributions; ii) *normality*. SBN-induced distributions are all naturally normalized if the associated conditional probability tables (CPTs) are consistent, which is a common property of Bayesian networks. The SBN framework also generalizes to unrooted trees, which are the most common type of trees in phylogenetics. Concretely, unrooted trees can be viewed as rooted trees with unobserved roots. Marginalizing out the unobserved root node $S_1$, we have the SBN probability estimates for unrooted trees

$$p_{\mathrm{sbn}}(T^{\mathrm{u}} = \tau) = \sum_{s_1 \sim \tau} p(S_1 = s_1) \prod_{i>1} p(S_i = s_i | S_{\pi_i} = s_{\pi_i})$$

where $\sim$ means all root subsplits that are compatible with $\tau$.

To reduce model complexity and encourage generalization, the same set of CPTs for parent-child subsplit pairs is shared across the SBN network, regardless of their locations. Similar to weight sharing used in convolutional networks (LeCun et al., 1998) for detecting *translationally-invariant structure* of images (e.g., edges, corners), this heuristic parameter sharing used in SBNs is for identifying *conditional splitting patterns* of phylogenetic trees. See Zhang & Matsen IV (2018) for more detailed discussion on SBNs.

## 3 VARIATIONAL PHYLOGENETIC INFERENCE VIA SBNS

The flexible and tractable tree topology distributions provided by SBNs serve as an essential building block to perform variational inference (Jordan et al., 1999) for phylogenetics. Suppose that we have a family of approximate distributions $Q_\phi(\tau)$ (e.g., SBNs) over phylogenetic tree topologies, where $\phi$ denotes the corresponding variational parameters (e.g., CPTs for SBNs). For each tree $\tau$, we posit another family of densities $Q_\psi(q|\tau)$ over the branch lengths, where $\psi$ is the branch length

variational parameters. We then combine these distributions and use the product

$$Q_{\phi,\psi}(\tau, \boldsymbol{q}) = Q_{\phi}(\tau)Q_{\psi}(\boldsymbol{q}|\tau)$$

as our variational approximation. Inference now amounts to finding the member of this family that minimizes the Kullback-Leibler (KL) divergence to the exact posterior,

$$\phi^*, \psi^* = \arg\min_{\phi,\psi} D_{KL}\left(Q_{\phi,\psi}(\tau, \boldsymbol{q}) \| p(\tau, \boldsymbol{q}|\boldsymbol{Y})\right) \tag{2}$$

which is equivalent to maximizing the evidence lower bound (ELBO),

$$L(\phi, \psi) = \mathbb{E}_{Q_{\phi,\psi}(\tau,\boldsymbol{q})} \log\left(\frac{p(\boldsymbol{Y}|\tau,\boldsymbol{q})p(\tau,\boldsymbol{q})}{Q_{\phi}(\tau)Q_{\psi}(\boldsymbol{q}|\tau)}\right) \leq \log p(\boldsymbol{Y}).$$

As the ELBO is based on a single-sample estimate of the evidence, it heavily penalizes samples that fail to explain the observed sequences. As a result, the variational approximation tends to cover only the high-probability areas of the true posterior. This effect can be minimized by averaging over $K > 1$ samples when estimating the evidence (Burda et al., 2016; Mnih & Rezende, 2016), which leads to tighter lower bounds

$$L^K(\phi, \psi) = \mathbb{E}_{Q_{\phi,\psi}(\tau^{1:K}, \boldsymbol{q}^{1:K})} \log\left(\frac{1}{K}\sum_{i=1}^{K}\frac{p(\boldsymbol{Y}|\tau^i,\boldsymbol{q}^i)p(\tau^i,\boldsymbol{q}^i)}{Q_{\phi}(\tau^i)Q_{\psi}(\boldsymbol{q}^i|\tau^i)}\right) \leq \log p(\boldsymbol{Y}) \tag{3}$$

where $Q_{\phi,\psi}(\tau^{1:K}, \boldsymbol{q}^{1:K}) \equiv \prod_{i=1}^{K} Q_{\phi,\psi}(\tau^i, \boldsymbol{q}^i)$; the tightness of the lower bounds improves as the number of samples $K$ increases (Burda et al., 2016). We will use multi-sample lower bounds in the sequel and refer to them as lower bounds for short.

### 3.1 VARIATIONAL PARAMETERIZATION

The CPTs in SBNs are, in general, associated with all possible parent-child subsplit pairs. Therefore, in principle a full parameterization requires an exponentially increasing number of parameters. In practice, however, we can find a sufficiently large subsplit *support* of CPTs (i.e. where the associated conditional probabilities are allowed to be nonzero) that covers favorable subsplit pairs from trees in the high-probability areas of the true posterior. In this paper, we will mostly focus on the variational approach and assume the support of CPTs is available, although in our experiments we find that a simple bootstrap-based approach does provide a reasonable CPT support estimate for real data. We leave the development of more sophisticated methods for finding the support of CPTs to future work.

Now denote the set of root subsplits in the support as $\mathbb{S}_r$ and the set of parent-child subsplit pairs in the support as $\mathbb{S}_{\mathrm{ch|pa}}$. The CPTs are defined according to the following equations

$$p(S_1 = s_1) = \frac{\exp(\phi_{s_1})}{\sum_{s_r \in \mathbb{S}_r} \exp(\phi_{s_r})}, \quad p(S_i = s|S_{\pi_i} = t) = \frac{\exp(\phi_{s|t})}{\sum_{s \in \mathbb{S}_{\cdot|t}} \exp(\phi_{s|t})}$$

where $\mathbb{S}_{\cdot|t}$ denotes the set of child subsplits for parent subsplit $t$.

We use the Log-normal distribution $\mathrm{Lognormal}(\mu, \sigma^2)$ as our variational approximation for branch lengths to accommodate their non-negative nature in phylogenetic models. Instead of a naive parameterization for each edge on each tree (which would require a large number of parameters when the high-probability areas of the posterior are diffuse), we use an amortized set of parameters over the shared local structures among trees. A simple choice of such local structures is the split, a bipartition $(X_1, X_2)$ of the leaf labels $\mathcal{X}$ (i.e. $X_1 \cup X_2 = \mathcal{X}, X_1 \cap X_2 = \emptyset$), and each edge of a phylogenetic tree naturally corresponds to a split, the bipartition that consists of the leaf labels from both sides of the edge. Note that a split can be viewed as a root subsplit. We then assign $\mu(\cdot, \cdot), \sigma(\cdot, \cdot)$ for each split $(\cdot, \cdot)$ in $\mathbb{S}_r$. We denote the corresponding split of edge $e$ of tree $\tau$ as $e/\tau$.

**A Simple Independent Approximation**   Given a phylogenetic tree $\tau$, we start with a simple model that assumes the branch lengths for the edges of the tree are independently distributed. The approximate density $Q_{\psi}(\boldsymbol{q}|\tau)$, therefore, has the form

$$Q_{\psi}(\boldsymbol{q}|\tau) = \prod_{e \in E(\tau)} p^{\mathrm{Lognormal}}\left(q_e \mid \mu(e, \tau), \sigma(e, \tau)\right), \quad \mu(e, \tau) = \psi_{e/\tau}^{\mu}, \ \sigma(e, \tau) = \psi_{e/\tau}^{\sigma}. \tag{4}$$

**Capturing Between-Tree Branch Length Variation**
The above approximation equation 4 implicitly assumes that the branch lengths in different trees have the same distribution if they correspond to the same split, which fails to account for between-tree variation. To capture this variation, one can use a more sophisticated parameterization that allows other tree-dependent terms for the variational parameters $\mu$ and $\sigma$. Specifically, we use additional local structure associated with each edge as follows:

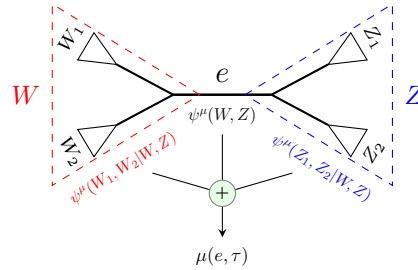

Figure 2: Branch length parameterization using primary subsplit pairs, which is the sum of parameters for a split and its neighboring subsplit pairs. Edge $e$ represents a split $(W, Z)$. Parameterization for the variance is the same as for the mean.

**Definition 1 (primary subsplit pair)** *Let $e$ be an edge of a phylogenetic tree $\tau$ which represents a split $e/\tau = (W, Z)$. Assume that at least one of $W$ or $Z$, say $W$, contains more than one leaf label and denote its subsplit as $(W_1, W_2)$. We call the parent-child subsplit pair $(W_1, W_2)|(W, Z)$ a* primary subsplit pair.

We assign additional parameters for each primary subsplit pair. Denoting the primary subsplit pair(s) of edge $e$ in tree $\tau$ as $e/\!/\tau$, we then simply sum all variational parameters associated with $e$ to form the mean and variance parameters for the corresponding branch length (Figure 2):

$$\mu(e, \tau) = \psi_{e/\tau}^{\mu} + \sum_{s \in e/\!/\tau} \psi_s^{\mu}, \quad \sigma(e, \tau) = \psi_{e/\tau}^{\sigma} + \sum_{s \in e/\!/\tau} \psi_s^{\sigma}.$$

This modifies the density in equation 4 by adding contributions from primary subsplit pairs and hence allows for more flexible between-tree approximations. Note that the above structured parameterizations of branch length distributions also enable joint learning across tree topologies.

## 3.2 Stochastic Gradient Estimators and The VBPI Algorithm

In practice, the lower bound is usually maximized via stochastic gradient ascent (SGA). However, the naive stochastic gradient estimator obtained by differentiating the lower bound has very large variance and is impractical for our purpose. Fortunately, various variance reduction techniques have been introduced in recent years including the *control variate* (Paisley et al., 2012; Ranganath et al., 2014; Mnih & Gregor, 2014; Mnih & Rezende, 2016) for general latent variables and the *reparameterization trick* (Kingma & Welling, 2014) for continuous latent variables. In the following, we apply these techniques to different components of our latent variables and derive efficient gradient estimators with much lower variance, respectively. In addition, we also consider a stable gradient estimator based on an alternative variational objective. See Appendix A for derivations.

**The VIMCO Estimator** Let $f_{\phi,\psi}(\tau, \boldsymbol{q}) = \frac{p(\boldsymbol{Y}|\tau,\boldsymbol{q})p(\tau,\boldsymbol{q})}{Q_{\phi}(\tau)Q_{\psi}(\boldsymbol{q}|\tau)}$. The stochastic lower bound with $K$ samples is $\hat{L}^K(\phi, \psi) = \log\left(\frac{1}{K}\sum_{i=1}^K f_{\phi,\psi}(\tau^i, \boldsymbol{q}^i)\right)$. Mnih & Rezende (2016) propose a localized learning signal strategy that significantly reduces the variance of the naive gradient estimator by utilizing the independence between the multiple samples and the regularity of the learning signal, which estimates the gradient as follows

$$\nabla_{\phi} L^K(\phi, \psi) = \mathbb{E}_{Q_{\phi,\psi}(\tau^{1:K}, \boldsymbol{q}^{1:K})} \sum_{j=1}^K \left(\hat{L}_{j|-j}^K(\phi, \psi) - \tilde{w}^j\right) \nabla_{\phi} \log Q_{\phi}(\tau^j) \tag{5}$$

where

$$\hat{L}_{j|-j}^K(\phi, \psi) := \hat{L}^K(\phi, \psi) - \log \frac{1}{K}\left(\sum_{i \neq j} f_{\phi,\psi}(\tau^i, \boldsymbol{q}^i) + \hat{f}_{\phi,\psi}(\tau^{-j}, \boldsymbol{q}^{-j})\right)$$

is the per-sample local learning signal, with $\hat{f}_{\phi,\psi}(\tau^{-j}, \boldsymbol{q}^{-j})$ being some estimate of $f_{\phi,\psi}(\tau^j, \boldsymbol{q}^j)$ for sample $j$ using the rest of samples (e.g., the geometric mean), and $\tilde{w}^j = \frac{f_{\phi,\psi}(\tau^j, \boldsymbol{q}^j)}{\sum_{i=1}^K f_{\phi,\psi}(\tau^i, \boldsymbol{q}^i)}$ is the self-normalized importance weight. This gives the following VIMCO estimator

$$\nabla_{\phi} L^K(\phi, \psi) \simeq \sum_{j=1}^K \left(\hat{L}_{j|-j}^K(\phi, \psi) - \tilde{w}^j\right) \nabla_{\phi} \log Q_{\phi}(\tau^j) \text{ with } \tau^j, \boldsymbol{q}^j \overset{\text{iid}}{\sim} Q_{\phi,\psi}(\tau, \boldsymbol{q}). \tag{6}$$

**The Reparameterization Trick**    The VIMCO estimator also works for the branch length gradient. However, as branch lengths are continuous latent variables, we can use the reparameterization trick to estimate the gradient. Because the Log-normal distribution has a simple reparameterization, $q \sim \text{Lognormal}(\mu, \sigma^2) \Leftrightarrow q = \exp(\mu + \sigma\epsilon), \ \epsilon \sim \mathcal{N}(0, 1)$, we can rewrite the lower bound:

$$L^K(\boldsymbol{\phi}, \boldsymbol{\psi}) = \mathbb{E}_{Q_{\boldsymbol{\phi},\boldsymbol{\epsilon}}(\tau^{1:K}, \boldsymbol{\epsilon}^{1:K})} \log \left( \frac{1}{K} \sum_{j=1}^{K} \frac{p(\boldsymbol{Y}|\tau^j, g_{\boldsymbol{\psi}}(\boldsymbol{\epsilon}^j|\tau^j)) p(\tau^j, g_{\boldsymbol{\psi}}(\boldsymbol{\epsilon}^j|\tau^j))}{Q_{\boldsymbol{\phi}}(\tau^j) Q_{\boldsymbol{\psi}}(g_{\boldsymbol{\psi}}(\boldsymbol{\epsilon}^j|\tau^j)|\tau^j)} \right).$$

where $g_{\boldsymbol{\psi}}(\boldsymbol{\epsilon}|\tau) = \exp(\boldsymbol{\mu}_{\boldsymbol{\psi},\tau} + \boldsymbol{\sigma}_{\boldsymbol{\psi},\tau} \odot \boldsymbol{\epsilon})$. Then the gradient of the lower bound w.r.t. $\boldsymbol{\psi}$ is

$$\nabla_{\boldsymbol{\psi}} L^K(\boldsymbol{\phi}, \boldsymbol{\psi}) = \mathbb{E}_{Q_{\boldsymbol{\phi},\boldsymbol{\epsilon}}(\tau^{1:K}, \boldsymbol{\epsilon}^{1:K})} \sum_{j=1}^{K} \tilde{w}^j \nabla_{\boldsymbol{\psi}} \log f_{\boldsymbol{\phi},\boldsymbol{\psi}}(\tau^j, g_{\boldsymbol{\psi}}(\boldsymbol{\epsilon}^j|\tau^j)) \tag{7}$$

where $\tilde{w}^j = \frac{f_{\boldsymbol{\phi},\boldsymbol{\psi}}(\tau^j, g_{\boldsymbol{\psi}}(\boldsymbol{\epsilon}^j|\tau^j))}{\sum_{i=1}^{K} f_{\boldsymbol{\phi},\boldsymbol{\psi}}(\tau^i, g_{\boldsymbol{\psi}}(\boldsymbol{\epsilon}^i|\tau^i))}$ is the same normalized importance weight as in equation equation 5. Therefore, we can form the Monte Carlo estimator of the gradient

$$\nabla_{\boldsymbol{\psi}} L^K(\boldsymbol{\phi}, \boldsymbol{\psi}) \simeq \sum_{j=1}^{K} \tilde{w}^j \nabla_{\boldsymbol{\psi}} \log f_{\boldsymbol{\phi},\boldsymbol{\psi}}(\tau^j, g_{\boldsymbol{\psi}}(\boldsymbol{\epsilon}^j|\tau^j)) \ \text{with} \ \tau^j \stackrel{\text{iid}}{\sim} Q_{\boldsymbol{\phi}}(\tau), \ \boldsymbol{\epsilon}^j \stackrel{\text{iid}}{\sim} \mathcal{N}(\boldsymbol{0}, \boldsymbol{I}). \tag{8}$$

**Self-normalized Importance Sampling Estimator**    In addition to the standard variational formulation equation 2, one can reformulate the optimization problem by minimizing the reversed KL divergence, which is equivalent to maximizing the likelihood of the variational approximation

$$Q_{\boldsymbol{\phi}^*,\boldsymbol{\psi}^*}(\tau, \boldsymbol{q}), \ \text{where} \ \boldsymbol{\phi}^*, \boldsymbol{\psi}^* = \underset{\boldsymbol{\phi},\boldsymbol{\psi}}{\arg\max} \ \tilde{L}(\boldsymbol{\phi}, \boldsymbol{\psi}), \ \tilde{L}(\boldsymbol{\phi}, \boldsymbol{\psi}) = \mathbb{E}_{p(\tau, \boldsymbol{q}|\boldsymbol{Y})} \log Q_{\boldsymbol{\phi},\boldsymbol{\psi}}(\tau, \boldsymbol{q}). \tag{9}$$

We can use an importance sampling estimator to compute the gradient of the objective

$$\nabla_{\boldsymbol{\phi}} \tilde{L}(\boldsymbol{\phi}, \boldsymbol{\psi}) = \mathbb{E}_{p(\tau, \boldsymbol{q}|\boldsymbol{Y})} \nabla_{\boldsymbol{\phi}} \log Q_{\boldsymbol{\phi}}(\tau) = \frac{1}{p(\boldsymbol{Y})} \mathbb{E}_{Q_{\boldsymbol{\phi},\boldsymbol{\psi}}(\tau, \boldsymbol{q})} \frac{p(\boldsymbol{Y}|\tau, \boldsymbol{q}) p(\tau, \boldsymbol{q})}{Q_{\boldsymbol{\phi}}(\tau) Q_{\boldsymbol{\psi}}(\boldsymbol{q}|\tau)} \nabla_{\boldsymbol{\phi}} \log Q_{\boldsymbol{\phi}}(\tau)$$

$$\simeq \sum_{j=1}^{K} \tilde{w}^j \nabla_{\boldsymbol{\phi}} \log Q_{\boldsymbol{\phi}}(\tau^j) \ \text{with} \ \tau^j, \boldsymbol{q}^j \stackrel{\text{iid}}{\sim} Q_{\boldsymbol{\phi},\boldsymbol{\psi}}(\tau, \ \boldsymbol{q}) \tag{10}$$

with the same importance weights $\tilde{w}^j$ as in equation 5. This can be viewed as a multi-sample generalization of the wake-sleep algorithm (Hinton et al., 1995) and was first used in the *reweighted wake-sleep* algorithm (Bornschein & Bengio, 2015) for training deep generative models. We therefore call the gradient estimator in equation 10 the RWS estimator. Like the VIMCO estimator, the RWS estimator also provides gradients for branch lengths. However, we find in practice that equation 8 that uses the reparameterization trick is more useful and often leads to faster convergence, although it uses a different optimization objective. A better understanding of this phenomenon would be an interesting subject of future research.

All stochastic gradient estimators introduced above can be used in conjunction with stochastic optimization methods such as SGA or some of its adaptive variants (e.g. Adam Kingma & Ba, 2015) to maximize the lower bounds. See algorithm 1 in Appendix B for a basic variational Bayesian phylogenetic inference (VBPI) approach.

## 4    EXPERIMENTS

Throughout this section we evaluate the effectiveness and efficiency of our variational framework for inference over phylogenetic trees. The simplest SBN (the one with a full and complete binary tree structure) is used for the phylogenetic tree topology variational distribution; we have found it to provide sufficiently accurate approximation. For real datasets, we estimate the CPT supports from ultrafast maximum likelihood phylogenetic bootstrap trees using UFBoot (Minh et al., 2013), which is a fast approximate bootstrap method based on efficient heuristics. We compare the performance of the VIMCO estimator and the RWS estimator with different variational parameterizations for the branch length distributions, while varying the number of samples in the training objective to

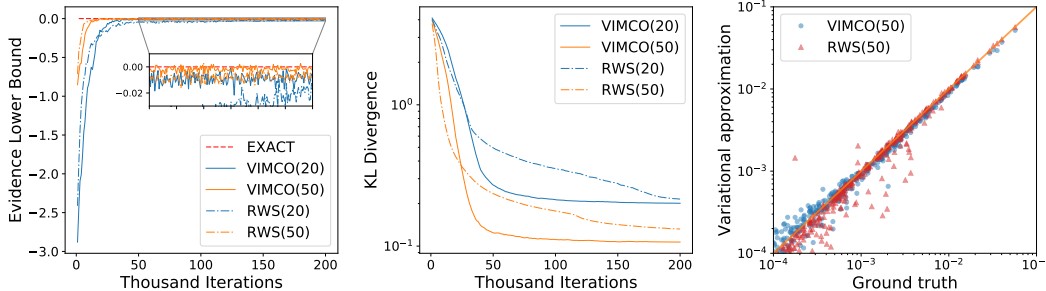

Figure 3: Comparison of multi-sample objective on approximating a challenging distribution over unrooted phylogenetic trees with 8 leaves using VIMCO and RWS gradient estimators. **Left:** Evidence lower bound. **Middle:** KL divergence. **Right:** Variational approximations vs ground truth probabilities. The number in brackets specifies the number of samples used in the training objective.

see how these affect the quality of the variational approximations. For VIMCO, we use Adam for stochastic gradient ascent with learning rate $0.001$ (Kingma & Ba, 2015). For RWS, we also use AMSGrad (Sashank et al., 2018), a recent variant of Adam, when Adam is unstable. Results were collected after 200,000 parameter updates. The KL divergences reported are over the discrete collection of phylogenetic tree structures, from trained SBN distribution to the ground truth, and a low KL divergence means a high quality approximation of the distribution of trees.

## 4.1 SIMULATED SCENARIOS

To empirically investigate the representative power of SBNs to approximate distributions on phylogenetic trees under the variational framework, we first conduct experiments on a simulated setup. We use the space of unrooted phylogenetic trees with 8 leaves, which contains 10395 unique trees in total. Given an arbitrary order of trees, we generate a target distribution $p_0(\tau)$ by drawing a sample from the symmetric Dirichlet distributions $\mathrm{Dir}(\beta\mathbf{1})$ of order 10395, where $\beta$ is the concentration parameter. The target distribution becomes more diffuse as $\beta$ increases; we used $\beta = 0.008$ to provide enough information for inference while allowing for adequate diffusion in the target. Note that there are no branch lengths in this simulated model and the lower bound is

$$L^K(\boldsymbol{\phi}) = \mathbb{E}_{Q_{\boldsymbol{\phi}}(\tau^{1:K})} \log \left( \frac{1}{K} \sum_{i=1}^{K} \frac{p_0(\tau^i)}{Q_{\boldsymbol{\phi}}(\tau^i)} \right) \leq 0$$

with the exact evidence being $\log(1) = 0$. We then use both the VIMCO and RWS estimators to optimize the above lower bound based on 20 and 50 samples ($K$). We use a slightly larger learning rate ($0.002$) in AMSGrad for RWS.

Figure 3 shows the empirical performance of different methods. From the left plot, we see that the lower bounds converge rapidly and the gaps between lower bounds and the exact model evidence are close to zero, demonstrating the expressive power of SBNs on phylogenetic tree probability estimations. The evolution of KL divergences (middle plot) is consistent with the lower bounds. All methods benefit from using more samples, with VIMCO performing better in the end and RWS learning slightly faster at the beginning.[1] The slower start of VIMCO is partly due to the regularization term in the lower bounds, which turns out to be beneficial for the overall performance since the regularization encourages the diversity of the variational approximation and leads to more sufficient exploration in the starting phase, similar to the exploring starts (ES) strategy in reinforcement learning (Sutton & Barto, 1998). The right plot compares the variational approximations obtained by VIMCO and RWS, both with 50 samples, to the ground truth $p_0(\tau)$. We see that VIMCO slightly underestimates trees in high-probability areas as a result of the regularization effect. While RWS provides better approximations for trees in high-probability areas, it tends to underestimate trees

---

[1]Although we use larger learning rates for RWS in this experiment, we found RWS generally learns slightly faster than VIMCO at the beginning. See Figure 4 for the real data phylogenetic inference problems in section 4.2 where we use Adam with learning rate 0.001 for both methods.

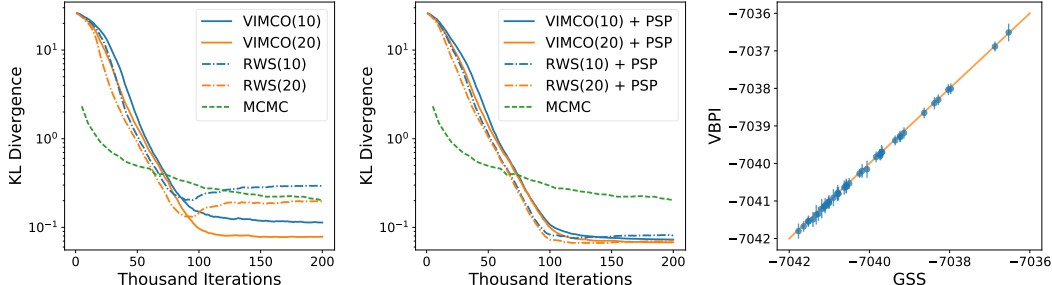

Figure 4: Performance on DS1. **Left:** KL divergence for methods that use the simple split-based parameterization for the branch length distributions. **Middle:** KL divergence for methods that use PSP. **Right:** Per-tree marginal likelihood estimation (in nats): VBPI vs GSS. The number in brackets specifies the number of samples used in the training objective. MCMC results are averaged over 10 independent runs. The results for VBPI were obtained using 1000 samples and the error bar shows one standard deviation over 100 independent runs.

in low-probability areas which deteriorates the overall performance. We expect the biases in both approaches to be alleviated with more samples.

### 4.2 REAL DATA PHYLOGENETIC POSTERIOR ESTIMATION

In the second set of experiments we evaluate the proposed variational Bayesian phylogenetic inference (VBPI) algorithms at estimating unrooted phylogenetic tree posteriors on 8 real datasets commonly used to benchmark phylogenetic MCMC methods (Lakner et al., 2008; Höhna & Drummond, 2012; Larget, 2013; Whidden & Matsen IV, 2015) (Table 1). We concentrate on the most challenging part of the phylogenetic model: joint learning of the tree topologies and the branch lengths. We assume a uniform prior on the tree topology, an i.i.d. exponential prior ($\mathrm{Exp}(10)$) for the branch lengths and the simple Jukes & Cantor (1969) substitution model. We consider two different variational parameterizations for the branch length distributions as introduced in section 3.1. In the first case, we use the simple split-based parameterization that assigns parameters to the splits associated with the edges of the trees. In the second case, we assign additional parameters for the primary subsplit pairs (PSP) to better capture the between-tree variation. We form our ground truth posterior from an extremely long MCMC run of 10 billion iterations (sampled each 1000 iterations with the first 25% discarded as burn-in) using MrBayes (Ronquist et al., 2012), and gather the support of CPTs from 10 replicates of 10000 ultrafast maximum likelihood bootstrap trees (Minh et al., 2013). Following Rezende & Mohamed (2015), we use a simple annealed version of the lower bound which was found to provide better results. The modified bound is:

$$L_{\beta_t}^K(\boldsymbol{\phi}, \boldsymbol{\psi}) = \mathbb{E}_{Q_{\boldsymbol{\phi},\boldsymbol{\psi}}(\tau^{1:K}, \, \boldsymbol{q}^{1:K})} \log \left( \frac{1}{K} \sum_{i=1}^{K} \frac{[p(\boldsymbol{Y}|\tau^i, \boldsymbol{q}^i)]^{\beta_t} p(\tau^i, \boldsymbol{q}^i)}{Q_{\boldsymbol{\phi}}(\tau^i) Q_{\boldsymbol{\psi}}(\boldsymbol{q}^i|\tau^i)} \right)$$

where $\beta_t \in [0, 1]$ is an inverse temperature that follows a schedule $\beta_t = \min(0.001, t/100000)$, going from 0.001 to 1 after 100000 iterations. We use Adam with learning rate 0.001 to train the variational approximations using VIMCO and RWS estimators with 10 and 20 samples.

Figure 4 (left and middle plots) shows the resulting KL divergence to the ground truth on DS1 as a function of the number of parameter updates. The results for methods that adopt the simple split-based parameterization of variational branch length distributions are shown in the left plot. We see that the performance of all methods improves significantly as the number of samples is increased. The middle plot, containing the results using PSP for variational parameterization, clearly indicates that a better modeling of between-tree variation of the branch length distributions is beneficial for all method / number of samples combinations. Specifically, PSP enables more flexible branch length distributions across trees which makes the learning task much easier, as shown by the considerably smaller gaps between the methods. To benchmark the learning efficiency of VBPI, we also compare to MrBayes 3.2.5 (Ronquist et al., 2012), a standard MCMC implementation. We run MrBayes with 4 chains and 10 runs for two million iterations, sampling every 100 iterations. For each run, we compute the KL divergence to the ground truth every 50000 iterations with the first 25% discarded

Table 1: Data sets used for variational phylogenetic posterior estimation, and marginal likelihood estimates of different methods across datasets. The marginal likelihood estimates of all variational methods are obtained by importance sampling using 1000 samples. We run stepping-stone in Mr-Bayes using default settings with 4 chains for 10,000,000 iterations and sampled every 100 iterations. The results are averaged over 10 independent runs with standard deviation in brackets.

| DATA SET | REFERENCE | (#TAXA, #SITES) | MARGINAL LIKELIHOOD (NATS) | | | | |
|---|---|---|---|---|---|---|---|
| | | | VIMCO(10) | VIMCO(20) | VIMCO(10) + PSP | VIMCO(20) + PSP | SS |
| DS1 | HEDGES ET AL. (1990) | (27, 1949) | -7108.43(0.26) | -7108.35(0.21) | -7108.41(0.16) | **-7108.42(0.10)** | -7108.42(0.18) |
| DS2 | GAREY ET AL. (1996) | (29, 2520) | -26367.70(0.12) | -26367.71(0.09) | **-26367.72(0.08)** | -26367.70(0.10) | -26367.57(0.48) |
| DS3 | YANG & YODER (2003) | (36, 1812) | -33735.08(0.11) | -33735.11(0.11) | **-33735.10(0.09)** | -33735.07(0.11) | -33735.44(0.50) |
| DS4 | HENK ET AL. (2003) | (41, 1137) | -13329.90(0.31) | -13329.98(0.20) | **-13329.94(0.18)** | -13329.93(0.22) | -13330.06(0.54) |
| DS5 | LAKNER ET AL. (2008) | (50, 378) | -8214.36(0.67) | -8214.74(0.38) | -8214.61(0.38) | -8214.55(0.43) | **-8214.51(0.28)** |
| DS6 | ZHANG & BLACKWELL (2001) | (50, 1133) | -6723.75(0.68) | -6723.71(0.65) | -6724.09(0.55) | **-6724.34(0.45)** | -6724.07(0.86) |
| DS7 | YODER & YANG (2004) | (59, 1824) | -37332.03(0.43) | -37331.90(0.49) | -37331.90(0.32) | **-37332.03(0.23)** | -37332.76(2.42) |
| DS8 | ROSSMAN ET AL. (2001) | (64, 1008) | -8653.34(0.55) | -8651.54(0.80) | **-8650.63(0.42)** | -8650.55(0.46) | -8649.88(1.75) |

as burn-in. For a relatively fair comparison (in terms of the number of likelihood evaluations), we compare 10 (i.e. 2·20/4) times the number of MCMC iterations with the number of 20-sample objective VBPI iterations.[2] Although MCMC converges faster at the start, we see that VBPI methods (especially those with PSP) quickly surpass MCMC and arrive at good approximations with much less computation. This is because VBPI iteratively updates the approximate distribution of trees (e.g., SBNs) which in turn allows guided exploration in the tree topology space. VBPI also provides the same majority-rule consensus tree as the ground truth MCMC run (Figure 5 in Appendix D).

The variational approximations provided by VBPI can be readily used to perform importance sampling for phylogenetic inference (more details in Appendix C). The right plot of Figure 4 compares VBPI using VIMCO with 20 samples and PSP to the state-of-the-art generalized stepping-stone (GSS) (Fan et al., 2011) algorithm for estimating the marginal likelihood of trees in the $95\%$ credible set of DS1. For GSS, we use 50 power posteriors and for each power posterior we run 1,000,000 MCMC iterations, sampling every 1000 iterations with the first $10\%$ discarded as burn-in. The reference distribution for GSS was obtained from an independent Gamma approximation using the maximum a posterior estimate. Table 1 shows the estimates of the marginal likelihood of the data (i.e., model evidence) using different VIMCO approximations and one of the state-of-the-art methods, the stepping-stone (SS) algorithm (Xie et al., 2011). For each data set, all methods provide estimates for the same marginal likelihood, with better approximation leading to lower variance. We see that VBPI using 1000 samples is already competitive with SS using 100000 samples and provides estimates with much less variance (hence more reproducible and reliable). Again, the extra flexibility enabled by PSP alleviates the demand for larger number of samples used in the training objective, making it possible to achieve high quality variational approximations with less samples.

## 5 DISCUSSION

In this work we introduced VBPI, a general variational framework for Bayesian phylogenetic inference. By combining subsplit Bayesian networks, a recent framework that provides flexible distributions of trees, and efficient structured parameterizations for branch length distributions, VBPI exhibits guided exploration (enabled by SBNs) in tree space and provides competitive performance to MCMC methods with less computation. Moreover, variational approximations provided by VBPI can be readily used for further statistical analysis such as marginal likelihood estimation for model comparison via importance sampling, which, compared to MCMC based methods, dramatically reduces the cost at test time. We report promising numerical results demonstrating the effectiveness and efficiency of VBPI on a benchmark of real data Bayesian phylogenetic inference problems.

When the data are weak and posteriors are diffuse, support estimation of CPTs becomes challenging. However, compared to classical MCMC approaches in phylogenetics that need to traverse the enormous support of posteriors on complete trees to accurately evaluate the posterior probabilities, the SBN parameterization in VBPI has a natural advantage in that it alleviates this issue by factorizing the uncertainty of complete tree topologies into local structures.

---

[2]the extra factor of 2/4 is because the likelihood and the gradient can be computed together in twice the time of a likelihood (Schadt et al., 1998) and we run MCMC with 4 chains.

Many topics remain for future work: constructing more flexible approximations for the branch length distributions (e.g., using normalizing flow (Rezende & Mohamed, 2015) for within-tree approximation and deep networks for the modeling of between-tree variation), deeper investigation of support estimation approaches in different data regimes, and efficient training algorithms for general variational inference on discrete / structured latent variables.

ACKNOWLEDGMENTS

This work supported by National Science Foundation grant CISE-1564137, as well as National Institutes of Health grant R01-GM113246. The research of Frederick Matsen was supported in part by a Faculty Scholar grant from the Howard Hughes Medical Institute and the Simons Foundation.

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

## A    GRADIENT DERIVATION FOR THE MULTI-SAMPLE OBJECTIVES

In this section we will derive the gradient for the multi-sample objectives introduced in section 3. We start with the lower bound

$$L^K(\boldsymbol{\phi}, \boldsymbol{\psi}) = \mathbb{E}_{Q_{\phi,\psi}(\tau^{1:K}, \, \boldsymbol{q}^{1:K})} \log \left( \frac{1}{K} \sum_{j=1}^{K} \frac{p(\boldsymbol{Y}|\tau^j, \boldsymbol{q}^j)p(\tau^j, \boldsymbol{q}^j)}{Q_{\phi}(\tau^j)Q_{\psi}(\boldsymbol{q}^j|\tau^j)} \right)$$

$$= \mathbb{E}_{Q_{\phi,\psi}(\tau^{1:K}, \, \boldsymbol{q}^{1:K})} \log \left( \frac{1}{K} \sum_{j=1}^{K} f_{\phi,\psi}(\tau^j, \boldsymbol{q}^j) \right).$$

Using the product rule and noting that $\nabla_{\phi} \log f_{\phi,\psi}(\tau^j, \boldsymbol{q}^j) = -\nabla_{\phi} \log Q_{\phi}(\tau^j)$,

$$\nabla_{\phi} L^k(\boldsymbol{\phi}, \boldsymbol{\psi}) = \mathbb{E}_{Q_{\phi,\psi}(\tau^{1:K}, \, \boldsymbol{q}^{1:K})} \nabla_{\phi} \log \left( \frac{1}{K} \sum_{j=1}^{K} f_{\phi,\psi}(\tau^j, \boldsymbol{q}^j) \right) +$$

$$\mathbb{E}_{Q_{\phi,\psi}(\tau^{1:K}, \, \boldsymbol{q}^{1:K})} \sum_{j=1}^{K} \frac{\nabla_{\phi} Q_{\phi}(\tau^j)}{Q_{\phi}(\tau^j)} \log \left( \frac{1}{K} \sum_{i=1}^{K} f_{\phi,\psi}(\tau^i, \boldsymbol{q}^i) \right)$$

$$= \mathbb{E}_{Q_{\phi,\psi}(\tau^{1:K}, \, \boldsymbol{q}^{1:K})} \sum_{j=1}^{K} \frac{f_{\phi,\psi}(\tau^j, \boldsymbol{q}^j)}{\sum_{i=1}^{K} f_{\phi,\psi}(\tau^i, \boldsymbol{q}^i)} \nabla_{\phi} \log f_{\phi,\psi}(\tau^j, \boldsymbol{q}^j) +$$

$$\mathbb{E}_{Q_{\phi,\psi}(\tau^{1:K}, \, \boldsymbol{q}^{1:K})} \sum_{j=1}^{K} \log \left( \frac{1}{K} \sum_{i=1}^{K} f_{\phi,\psi}(\tau^i, \boldsymbol{q}^i) \right) \nabla_{\phi} \log Q_{\phi}(\tau^j)$$

$$= \mathbb{E}_{Q_{\phi,\psi}(\tau^{1:K}, \, \boldsymbol{q}^{1:K})} \sum_{j=1}^{K} \left( \hat{L}^K(\boldsymbol{\phi}, \boldsymbol{\psi}) - \tilde{w}^j \right) \nabla_{\phi} \log Q_{\phi}(\tau^j).$$

This gives the naive gradient of the lower bound w.r.t. $\phi$.

Using the reparameterization trick, the lower bound has the form

$$L^K(\boldsymbol{\phi}, \boldsymbol{\psi}) = \mathbb{E}_{Q_{\phi,\epsilon}(\tau^{1:K}, \epsilon^{1:K})} \log \left( \frac{1}{K} \sum_{j=1}^{K} \frac{p(\boldsymbol{Y}|\tau^j, g_{\psi}(\epsilon^j|\tau^j))p(\tau^j, g_{\psi}(\epsilon^j|\tau^j))}{Q_{\phi}(\tau^j)Q_{\psi}(g_{\psi}(\epsilon^j|\tau^j)|\tau^j)} \right)$$

$$= \mathbb{E}_{Q_{\phi,\epsilon}(\tau^{1:K}, \epsilon^{1:K})} \log \left( \frac{1}{K} \sum_{j=1}^{K} f_{\phi,\psi}(\tau^j, g_{\psi}(\epsilon^j|\tau^j)) \right)$$

Since $\psi$ is not involved in the distribution with respect to which we take expectation,

$$\nabla_{\psi} L^K(\boldsymbol{\phi}, \boldsymbol{\psi}) = \mathbb{E}_{Q_{\phi,\epsilon}(\tau^{1:K}, \epsilon^{1:K})} \nabla_{\psi} \log \left( \frac{1}{K} \sum_{j=1}^{K} f_{\phi,\psi}(\tau^j, g_{\psi}(\epsilon^j|\tau^j)) \right)$$

$$= \mathbb{E}_{Q_{\phi,\epsilon}(\tau^{1:K}, \epsilon^{1:K})} \sum_{j=1}^{K} \frac{f_{\phi,\psi}(\tau^j, g_{\psi}(\epsilon^j|\tau^j))}{\sum_{i=1}^{K} f_{\phi,\psi}(\tau^i, g_{\psi}(\epsilon^i|\tau^i))} \nabla_{\psi} \log f_{\phi,\psi}(\tau^j, g_{\psi}(\epsilon^j|\tau^j))$$

$$= \mathbb{E}_{Q_{\phi,\epsilon}(\tau^{1:K}, \epsilon^{1:K})} \sum_{j=1}^{K} \tilde{w}^j \nabla_{\psi} \log f_{\phi,\psi}(\tau^j, g_{\psi}(\epsilon^j|\tau^j)).$$

Next, we derive the gradient of the multi-sample likelihood objective used in RWS

$$\tilde{L}(\boldsymbol{\phi}, \boldsymbol{\psi}) = \mathbb{E}_{p(\tau, \boldsymbol{q}|\boldsymbol{Y})} \log Q_{\phi,\psi}(\tau, \boldsymbol{q}).$$

Again, $p(\tau, \boldsymbol{q}|\boldsymbol{Y})$ is independent of $\boldsymbol{\phi}, \boldsymbol{\psi}$, and we have

$$
\begin{aligned}
\nabla_{\boldsymbol{\phi}} \tilde{L}(\boldsymbol{\phi}, \boldsymbol{\psi}) &= \mathbb{E}_{p(\tau, \boldsymbol{q}|\boldsymbol{Y})} \nabla_{\boldsymbol{\phi}} \log Q_{\boldsymbol{\phi}, \boldsymbol{\psi}}(\tau, \boldsymbol{q}) \\
&= \mathbb{E}_{Q_{\boldsymbol{\phi}, \boldsymbol{\psi}}(\tau, \boldsymbol{q})} \frac{p(\tau, \boldsymbol{q}|\boldsymbol{Y})}{Q_{\boldsymbol{\phi}}(\tau) Q_{\boldsymbol{\psi}}(\boldsymbol{q}|\tau)} \nabla_{\boldsymbol{\phi}} \log Q_{\boldsymbol{\phi}, \boldsymbol{\psi}}(\tau, \boldsymbol{q}) \\
&= \frac{1}{p(\boldsymbol{Y})} \mathbb{E}_{Q_{\boldsymbol{\phi}, \boldsymbol{\psi}}(\tau, \boldsymbol{q})} f_{\boldsymbol{\phi}, \boldsymbol{\psi}}(\tau, \boldsymbol{q}) \nabla_{\boldsymbol{\phi}} \log Q_{\boldsymbol{\phi}}(\tau) \\
&\simeq \sum_{j=1}^{K} \frac{f_{\boldsymbol{\phi}, \boldsymbol{\psi}}(\tau^j, \boldsymbol{q}^j)}{\sum_{i=1}^{k} f_{\boldsymbol{\phi}, \boldsymbol{\psi}}(\tau^i, \boldsymbol{q}^i)} \nabla_{\boldsymbol{\phi}} \log Q_{\boldsymbol{\phi}}(\tau^j) \quad \text{with} \quad \tau^j, \boldsymbol{q}^j \stackrel{\text{iid}}{\sim} Q_{\boldsymbol{\phi}, \boldsymbol{\psi}}(\tau, \boldsymbol{q}). \\
&= \sum_{j=1}^{K} \tilde{w}^j \nabla_{\boldsymbol{\phi}} \log Q_{\boldsymbol{\phi}}(\tau^j)
\end{aligned}
$$

The second to last step uses self-normalized importance sampling with $K$ samples. $\nabla_{\boldsymbol{\psi}} \tilde{L}(\boldsymbol{\phi}, \boldsymbol{\psi})$ can be computed in a similar way.

## B  THE VARIATIONAL BAYESIAN PHYLOGENETIC INFERENCE ALGORITHM

---

**Algorithm 1** The variational Bayesian phylogenetic inference (VBPI) algorithm.

---

1: $\boldsymbol{\phi}, \boldsymbol{\psi} \leftarrow$ Initialize parameters
2: **while** not converged **do**
3:    $\tau^1, \ldots, \tau^K \leftarrow$ Random samples from the current approximating tree distribution $Q_{\boldsymbol{\phi}}(\tau)$
4:    $\boldsymbol{\epsilon}^1, \ldots, \boldsymbol{\epsilon}^K \leftarrow$ Random samples from the multivariate standard normal distribution $\mathcal{N}(\boldsymbol{0}, \boldsymbol{I})$
5:    $\boldsymbol{g} \leftarrow \nabla_{\boldsymbol{\phi}, \boldsymbol{\psi}} L^K(\boldsymbol{\phi}, \boldsymbol{\psi}; \tau^{1:K}, \boldsymbol{\epsilon}^{1:K})$ (Use any gradient estimator from section 3.2)
6:    $\boldsymbol{\phi}, \boldsymbol{\psi} \leftarrow$ Update parameters using gradients $\boldsymbol{g}$ (e.g. SGA)
7: **end while**
8: **return** $\boldsymbol{\phi}, \boldsymbol{\psi}$

---

## C  IMPORTANCE SAMPLING FOR PHYLOGENETIC INFERENCE VIA VARIATIONAL APPROXIMATIONS

In this section, we provide a detailed importance sampling procedure for marginal likelihood estimation for phylogenetic inference based on the variational approximations provided by VBPI.

### C.1  ESTIMATING MARGINAL LIKELIHOOD OF TREES

For each tree $\tau$ that is covered by the subsplit support,

$$
Q_{\boldsymbol{\psi}}(\boldsymbol{q}|\tau) = \prod_{e \in E(\tau)} p^{\text{Lognormal}}(q_e \mid \mu(e, \tau), \sigma(e, \tau))
$$

can provide accurate approximation to the posterior of branch lengths on $\tau$, where the mean and variance parameters $\mu(e, \tau), \sigma(e, \tau)$ are gathered from the structured variational parameters $\boldsymbol{\psi}$ as introduced in section 3.1. Therefore, we can estimate the marginal likelihood of $\tau$ using importance sampling with $Q_{\boldsymbol{\psi}}(\boldsymbol{q}|\tau)$ being the importance distribution as follows

$$
p(\boldsymbol{Y}|\tau) = \mathbb{E}_{Q_{\boldsymbol{\psi}}(\boldsymbol{q}|\tau)} \frac{p(\boldsymbol{Y}|\tau, \boldsymbol{q}) p(\boldsymbol{q})}{Q_{\boldsymbol{\psi}}(\boldsymbol{q}|\tau)} \simeq \frac{1}{M} \sum_{j=1}^{M} \frac{p(\boldsymbol{Y}|\tau, \boldsymbol{q}^j) p(\boldsymbol{q}^j)}{Q_{\boldsymbol{\psi}}(\boldsymbol{q}^j|\tau)} \quad \text{with } \boldsymbol{q}^j \stackrel{\text{iid}}{\sim} Q_{\boldsymbol{\psi}}(\boldsymbol{q}|\tau)
$$

### C.2  ESTIMATING MODEL EVIDENCE

Similarly, we can estimate the marginal likelihood of the data as follows

$$
p(\boldsymbol{Y}) = \mathbb{E}_{Q_{\boldsymbol{\phi}, \boldsymbol{\psi}}}(\tau, \boldsymbol{q}) \frac{p(\boldsymbol{Y}|\tau, \boldsymbol{q}) p(\tau, \boldsymbol{q})}{Q_{\boldsymbol{\phi}}(\tau) Q_{\boldsymbol{\psi}}(\boldsymbol{q}|\tau)} \simeq \frac{1}{K} \sum_{j=1}^{K} \frac{p(\boldsymbol{Y}|\tau^j, \boldsymbol{q}^j) p(\tau^j, \boldsymbol{q}^j)}{Q_{\boldsymbol{\phi}}(\tau^j) Q_{\boldsymbol{\psi}}(\boldsymbol{q}^j|\tau^j)} \quad \text{with } \tau^j, \boldsymbol{q}^j \stackrel{\text{iid}}{\sim} Q_{\boldsymbol{\phi}, \boldsymbol{\psi}}(\tau, \boldsymbol{q}).
$$

In our experiments, we use $K = 1000$. When taking a log transformation, the above Monte Carlo estimate is no longer unbiased (for the evidence $\log p(\boldsymbol{Y})$). Instead, it can be viewed as one sample Monte Carlo estimate of the lower bound

$$L^K(\boldsymbol{\phi}, \boldsymbol{\psi}) = \mathbb{E}_{Q_{\boldsymbol{\phi},\boldsymbol{\psi}}(\tau^{1:K}, \boldsymbol{q}^{1:K})} \log \left( \frac{1}{K} \sum_{i=1}^{K} \frac{p(\boldsymbol{Y}|\tau^i, \boldsymbol{q}^i)p(\tau^i, \boldsymbol{q}^i)}{Q_{\boldsymbol{\phi}}(\tau^i)Q_{\boldsymbol{\psi}}(\boldsymbol{q}^i|\tau^i)} \right) \leq \log p(\boldsymbol{Y}) \qquad (11)$$

whose tightness improves as the number of samples $K$ increases. Therefore, with a sufficiently large $K$, we can use the lower bound estimate as a proxy for Bayesian model selection.

## D    CONSENSUS TREE COMPARISON ON DS1

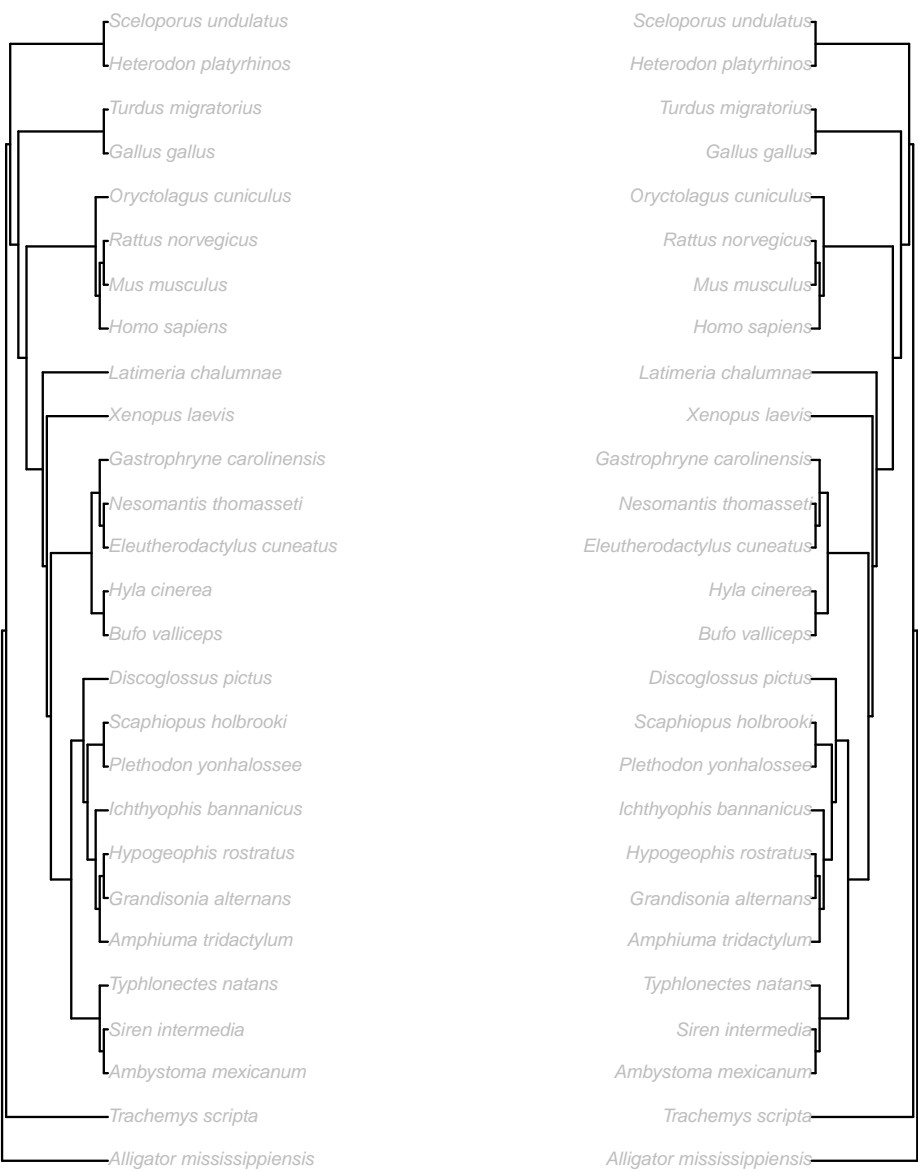

Figure 5: A comparison of majority-rule consensus trees obtained from VBPI and ground truth MCMC run on DS1. **Left:** Ground truth MCMC. **Right:** VBPI (10000 sampled trees). The plot is created using the *treespace* (Jombart et al., 2017) R package.

