# OpenReview forum: "Variational Bayesian Phylogenetic Inference"
_ICLR.cc/2019/Conference_

### Official Review · AnonReviewer3 · 2018-11-05
**A novel well executed paper**

**Rating:** 7
**Confidence:** 1

**Review:**

This paper is well written, appears to be well executed, and the results look good. I am not particularly well informed about the area, but the work appears to be novel. MCMC for phylogenetic inference is hugely expensive, and anything we can do to reduce that cost would be beneficial (the computational expense is not given, or I've missed it, for the variational approach - presumably it is relatively small compared to MCMC?).

My main criticism is that I found the details of subsplit Bayesian networks difficult to follow. Googling them suggests they are a relatively new model, which has not been well studied or used (there are no citations of the paper that introduces them for example!). The paper would be stronger if it discussed these in more detail - how close can they come to approximating the models usually used in phylogenetic analyses? Often the inferred phylogeny is itself of interest - how similar are the trees inferred here to those found from MrBayes?

---

> ### Author Response · Authors · 2018-11-13
> **Computational expense and approximation performance compared to MrBayes were reported.**
>
> Thank you for your review and feedback. We address your specific questions and comments below:
>
> 1) "the computational expense is not given, or I've missed it, for the variational approach - presumably it is relatively small compared to MCMC?"
>
> We present the computational expense for the variational approach in terms of the number of likelihood evaluations, and compare to MCMC. We direct the reviewer to Figure 4 in section 4.2, where we show the KL divergence to the ground truth as a function of the number of iterations of different methods (including MCMC via MrBayes). For a fair comparison, the number of iterations for MCMC is mapped to the number of iterations of variational methods that take the same number of likelihood evaluations.
>
> 2) "My main criticism is that I found the details of subsplit Bayesian networks difficult to follow. Googling them suggests they are a relatively new model, which has not been well studied or used (there are no citations of the paper that introduces them for example!)."
>
> SBNs are indeed a new model. The relatively short discussion of subsplit Bayesian networks (SBNs) is mainly due to the page limit of the conference, but we would like to present a more detailed discussion of SBNs if there is room in our revision. For a more detailed discussion, we refer the reviewer to the original paper [1] that introduced SBNs, which has been accepted to NIPS this year.
>
> 3) "The paper would be stronger if it discussed these in more detail - how close can they come to approximating the models usually used in phylogenetic analyses? Often the inferred phylogeny is itself of interest - how similar are the trees inferred here to those found from MrBayes?"
>
> First, we would like to ensure that our means of evaluating the SBN approximation is clear. We compute KL divergence over the discrete collection of phylogenetic tree structures, from the SBN distribution to the ground truth distribution on phylogenetic tree models obtained from extremely long MCMC runs using MrBayes. In order to get a low KL divergence to this ground truth, it is not enough to have similar trees: one must find practically the same set of trees as MrBayes, with nearly identical probability weights.
>
> Based on the low KL divergence reported in [1] and our experiments, SBNs can indeed provide accurate approximations to the phylogenetic posteriors inferred from real data (see Table 1 in [1] and section 4.2 in our paper.). Therefore, we believe SBN-based phylogenetic inference represents an important advance in this field, especially on structural learning of phylogenies.
>
>
> Reference
> [1] C. Zhang and FA. Matsen. Generalizing tree probability estimation via Bayesian networks. arXiv preprint arXiv:1805.07834, 2018

---

### Official Review · AnonReviewer2 · 2018-11-06
**A nice approach to inferring phylogenetic trees**

**Rating:** 5
**Confidence:** 3

**Review:**

This paper proposes a variational approach to Bayesian posterior inference in phylogenetic trees. The novel part of the approach (using subsplit Bayesian networks as a variational distribution) is intelligently combined with recent ideas from the approximate-inference literature (reweighted wake-sleep, VIMCO, reparameterization gradients, and multiple-sample ELBO estimators) to yield what seems to be an effective approach to a very hard inference problem.

My score would be higher were it not for two issues:
* The paper is 10 pages long, and I'm not convinced it needs to be. The reviewer guidelines (https://iclr.cc/Conferences/2019/Reviewer_Guidelines) say that "the overall time to read a paper should be comparable to that of a typical 8-page conference paper. Reviewers may apply a higher reviewing standard to papers that substantially exceed this length." So I recommend trying to cut it down a bit during the revision phase.
* The empirical comparisons are all likelihood/ELBO-based. These metrics are important, but it would be nice to see some kind of qualitative summary of the inferences made by different methods—two methods can produce similar log-likelihoods or KL divergences but suggest different scientific conclusions.

One final comment: it's not clear to me that ICLR is the most relevant venue for this work, which is purely about Bayesian inference rather than deep learning. This isn't a huge deal—certainly there's plenty of variational inference at ICLR these days—but I suspect many ICLR attendees may tune out when they realize there aren't any neural nets in the paper.

---

> ### Author Response · Authors · 2018-11-13
> **Thanks for the suggestions and we need some clarifications on your part**
>
> We thank the reviewer for your review and time. We would like to incorporate the suggestions into our revision and think we would benefit from some clarifications on your part.
>
> 1) "The paper is 10 pages long, and I'm not convinced it needs to be. The reviewer guidelines (https://iclr.cc/Conferences/2019/Reviewer_Guidelines) say that "the overall time to read a paper should be comparable to that of a typical 8-page conference paper. Reviewers may apply a higher reviewing standard to papers that substantially exceed this length." So I recommend trying to cut it down a bit during the revision phase."
>
> The main reason we took 10 pages for the paper is that phylogenetic inference is probably not well known to the machine learning community and much space is devoted to putting the phylogenetic models and experiments in context. We have tried to balance between being short and being a little bit long (but more self-contained) and thought the latter would eventually save the reviewers' time. However, we would like to cut down our paper as suggested and would appreciate it very much if the reviewer can point to us which parts of the paper that you find are redundant and can be made more brief.
>
> 2) "The empirical comparisons are all likelihood/ELBO-based. These metrics are important, but it would be nice to see some kind of qualitative summary of the inferences made by different methods?two methods can produce similar log-likelihoods or KL divergences but suggest different scientific conclusions."
>
> First, we would like to make sure the reviewer is aware how the KL results show the SBN-based approximations to be very close in distribution on the discrete space of phylogenetic tree structures. We have emphasized in point 3 to reviewer 3, and realize that we should have been more clear on this point.
>
> However, we are happy to incorporate any qualitative summaries the reviewer would like to suggest. We could certainly add, for example, tree shape summaries, but such a comparison would be significantly weaker than the current comparison on tree structures.
>
> 3) "One final comment: it's not clear to me that ICLR is the most relevant venue for this work, which is purely about Bayesian inference rather than deep learning. This isn't a huge deal?certainly there's plenty of variational inference at ICLR these days?but I suspect many ICLR attendees may tune out when they realize there aren't any neural nets in the paper."
>
> We think ICLR is an excellent venue for this work because: (i) Representation learning on discrete/structured objects has received increasing attention from the machine learning community, and our work represents an important advance in variational inference on complex structured models. (ii) Our variational framework admits many extensions that can incorporate the approximating power of neural networks (e.g, using normalizing flow and deep networks for more flexible within-tree and between-tree approximations, as mentioned in the discussion section of our paper).

---

### Official Review · AnonReviewer4 · 2018-11-12
**New approximate inference approaches for phylogenetic trees**

**Rating:** 6
**Confidence:** 3

**Review:**

This paper explores an approximate inference solution to the challenging problem of Bayesian inference of phylogenetic trees. Its leverages recently proposed subsplit Bayesian networks (SBNs) as a variational approximation over tree space and combines this with modern gradient estimators for VI. It is thorough in its evaluation of both methodological considerations and different datasets.

The main advantage would seem to be a large speedup over MCMC-based methods (Figure 4), which could be of significant value to the phylogenetics community. This point would benefit from more discussion. How do the number of iterations (reported in Figures 3&4, which was done carefully) correspond to wallclock time? Can this new method scale to numbers of sites and sequences that were previously unfeasible?

The main technical contribution is the use of SBNs as variational approximations over tree-space, but it is difficult to follow their implementation and parameter sharing without the explanation of the original paper. Additionally, the issue of estimating the support of the subsplit CPTs needs more discussion. As the authors acknowledge, complete parameterizations of these models scale in a combinatorial way with “all possible parent-child subsplit pairs”, and they deal with this by shrinking the support up front with various heuristics. It seems that these support estimation approaches would be feasible when the data are strong but would become challenging to scale when the data are weak. Since VB is often concerned with the limited-data regime, more discussion of when support estimation is feasible and when it is difficult would clarify how widely applicable the method is.

Overall, this work is an interesting extension of variational Bayes to a tree-structured inference problem and is thorough in its evaluation. While it is a bit focused on classical inference for ICLR, it could be interesting both for the VI community and as a significant application advancement.

Other notes:
In table 1, is the point that all methods are basically the same with different variance? This is not clear from the text. What about the variational bounds?

---

> ### Author Response · Authors · 2018-11-14
> **Clarification on technical issues  (Part 1/2)**
>
> Thank you for your thoughtful review and valuable feedback. Below are the answers to your comments:
>
> 1) "The main advantage would seem to be a large speedup over MCMC-based methods (Figure 4), which could be of significant value to the phylogenetics community. This point would benefit from more discussion. How do the number of iterations (reported in Figures 3&4, which was done carefully) correspond to wallclock time? Can this new method scale to numbers of sites and sequences that were previously unfeasible?"
>
> We are glad that this reviewer appreciates the care with which we crafted the comparison in terms of number of likelihood evaluations. Our motivation in doing a comparison in terms of likelihood evaluations is because our current implementation is in Python, whereas while MrBayes is in C that has been optimized for many years. This is the first paper introducing the ideas and initial implementation of variational Bayes phylogenetic inference, and we think that this level of comparison is appropriate. We will soon begin developing a highly optimized implementation, for which we are planning a more applications-driven paper which will include a wallclock comparison.
>
> Regarding large data sets, given that the learned SBNs can provide guided exploration in tree space and variational approaches naturally incorporate stochastic gradients, we believe it is much easier for VBPI to scale to datasets with large numbers of sequences and sites. However, we have not tried out our initial Python implementation on especially big data sets.
>
> 2) "The main technical contribution is the use of SBNs as variational approximations over tree-space, but it is difficult to follow their implementation and parameter sharing without the explanation of the original paper."
>
> As explained in point 2 to reviewer 3, this is mainly due to the page limit of the conference. We will definitely add more detailed explanation in our revision if there is room after trimming proposed by Reviewer 2.
>
> 3) "Additionally, the issue of estimating the support of the subsplit CPTs needs more discussion. As the authors acknowledge, complete parameterizations of these models scale in a combinatorial way with ?all possible parent-child subsplit pairs?, and they deal with this by shrinking the support up front with various heuristics. It seems that these support estimation approaches would be feasible when the data are strong but would become challenging to scale when the data are weak. Since VB is often concerned with the limited-data regime, more discussion of when support estimation is feasible and when it is difficult would clarify how widely applicable the method is."
>
> This is indeed an important point. We agree that when the data are weak, the posterior on subsplit pairs could have a large support.
>
> However, the SBN approach actually has a strong natural advantage in the weak-data regime. When data is weak, the support of the posterior distribution on complete trees, as evaluated by classical MCMC approaches, is enormous. For example, if there is uncertainty in multiple different parts of the tree, the support on complete trees scales as the product of these local uncertainties.
>
> The SBN parameterization alleviates this issue by factorizing the uncertainty into local structures. Thus, if the support of parent-child pairs is too large, then one should certainly not be trying to assign posterior support to each tree individually as in classical MCMC.
>
> Regarding heuristics for support estimation, we show in section 4.2 that bootstrap-based support estimation is effective even for diffuse posteriors across four data sets (DS5, DS6, DS7, DS8). See below for the numbers of unique trees in the standard MCMC run samples for all data sets (which is an indicator of the diffusivity of the posteriors).
>
> -------------------------------------------------------------------------------------------------------------------------
> datasets                |     DS1        DS2        DS3         DS4         DS5          DS6         DS7        DS8
> -------------------------------------------------------------------------------------------------------------------------
> # sample trees     |    1228          7           43           828        33752      35407       1125      3067
> -------------------------------------------------------------------------------------------------------------------------
>
> We agree that a further discussion of the weak-data regime is important and we look forward to adding to the discussion in a revision.

---

> ### Author Response · Authors · 2018-11-14
> **Clarification on technical issues (Part 2/2)**
>
> 4) "In table 1, is the point that all methods are basically the same with different variance? This is not clear from the text. What about the variational bounds?"
>
> Yes, you are right. All methods provide estimates for the same marginal likelihood, and better approximation would lead to smaller variance. The phylogenetic model is well defined with fixed structure, in contrast to generative models (e.g., VAE) where the generative network is trainable. In the paper we do not report the variational bounds since we want to compare to the stepping-stone (SS) algorithm on marginal likelihood estimation. These lower bounds are definitely different and improve as more particles and more flexible approximations are adopted, which we list below (averaged over 1000 runs).
>
> In our revision we look forward to clarifying this point.
>
>                                            Variational Lower Bounds
> +---------+------------------+------------------+--------------------------+--------------------------+
> |	      |  VIMCO(10)  |  VIMCO(20)  |   VIMCO(10)+PSP   |   VIMCO(20)+PSP   |
> +---------+------------------+------------------+--------------------------+--------------------------+
> |  DS1   |    -7108.91    |    -7108.77     |         -7108.73          |          -7108.61         |
> +---------+------------------+------------------+--------------------------+--------------------------+
> |  DS2   |   -26367.91   |    -26367.82   |         -26367.89       |          -26367.83       |
> +---------+------------------+------------------+--------------------------+--------------------------+
> |  DS3   |   -33735.36   |    -33735.26   |         -33735.29       |          -33735.24       |
> +---------+------------------+------------------+--------------------------+--------------------------+
> |  DS4   |   -13330.49   |    -13330.32   |         -13330.37       |          -13330.22       |
> +---------+------------------+------------------+--------------------------+--------------------------+
> |  DS5   |   -8215.85     |    -8215.56     |         -8215.64          |         -8215.36          |
> +---------+------------------+------------------+--------------------------+--------------------------+
> |  DS6   |   -6725.69     |    -6725.42     |         -6725.48          |         -6725.19          |
> +---------+------------------+------------------+--------------------------+--------------------------+
> |  DS7   |   -37332.91   |    -37332.65   |         -37332.72       |          -37332.49       |
> +---------+------------------+------------------+--------------------------+--------------------------+
> |  DS8   |   -8655.02     |    -8652.55     |         -8651.76          |         -8651.53          |
> +---------+------------------+------------------+--------------------------+--------------------------+

---

### Author Response · Authors · 2018-11-13
**A note to all reviewers**

We thank the reviewers for their reviews and the time spent on the manuscript, and their encouragement on the novelty of our work. We want to emphasize that, in addition to using subsplit Bayesian networks for approximating phylogenetic tree posteriors, our structured parameterization of the branch length distributions is also novel which allows us to jointly learn the branch length distributions across tree topologies. In contrast, classical MCMC typically uses simple random perturbations which contributes to the low acceptance rate for large topological modifications.

---

### Author Response · Authors · 2018-11-21
**Revision summary**

We thank all reviewers for the constructive feedback. We have revised the paper, and have incorporated their suggestions with the following major changes:

- We reorganized the SBN section and added more detailed discussion on SBN implementations and parameter sharing to better explain the subsplit Bayesian network framework.

- We cut down the paper a bit, in particular in the stochastic gradient estimator section, and now the main text (not including the discussion) is within 9 pages.

- We added more discussion on support estimation feasibility to the discussion section.

- We clarified a bit the KL divergence used in our experiments, and explained why a low KL divergence is a really strong statement about the quality of the tree posterior approximation. We hope that this addresses the concerns of reviewer 3 about the approximation performance on tree topologies.

- We added an additional qualitative summary (a consensus tree) of VBPI and MCMC to our experiments (Figure 5 in Appendix D). This tree shows that the MCMC-inferred and the VB-inferred consensus trees are identical. We hope that this addresses the concerns of reviewers 2 and 3 who were interested in more qualitative and scientifically-relevant posterior summaries.

- We clarified a bit the marginal likelihood estimates presented in Table 1, more clearly describing why VBPI provides a substantial advantage for this task.

We hope our revision has adequately addressed the reviewers' questions and concerns, and look forward to reading any other additional comments.

---

### Meta-Review · Area_Chair1 · 2018-12-14
**Meta-Review for Phylogenetic Inference paper**

**Confidence:** 3
**Recommendation:** Accept (Poster)

**Metareview:**

The reviewers lean to accept, and the authors clearly put a significant amount of time into their response. I will also lean to accept. However, the comments of reviewer 2 should be taken seriously, and addressed if possible, including an attempt to cut the paper length down.